# Universal Humanoid Motion Representations for Physics-Based Control

Zhengyi Luo[1,2]   Jinkun Cao[2]   Josh Merel[1]   Alexander Winkler[1]   Jing Huang[1]
Kris Kitani[1,2] *   Weipeng Xu[1] *

[1]Reality Labs Research, Meta; [2]Carnegie Mellon University
https://zhengyiluo.github.io/PULSE/

## Abstract

We present a universal motion representation that encompasses a comprehensive range of motor skills for physics-based humanoid control. Due to the high dimensionality of humanoids and the inherent difficulties in reinforcement learning, prior methods have focused on learning skill embeddings for a narrow range of movement styles (*e.g.* locomotion, game characters) from specialized motion datasets. This limited scope hampers their applicability in complex tasks. We close this gap by significantly increasing the coverage of our motion representation space. To achieve this, we first learn a motion imitator that can imitate *all* of human motion from a large, unstructured motion dataset. We then create our motion representation by distilling skills directly from the imitator. This is achieved by using an encoder-decoder structure with a variational information bottleneck. Additionally, we jointly learn a prior conditioned on proprioception (humanoid's own pose and velocities) to improve model expressiveness and sampling efficiency for downstream tasks. By sampling from the prior, we can generate long, stable, and diverse human motions. Using this latent space for hierarchical RL, we show that our policies solve tasks using human-like behavior. We demonstrate the effectiveness of our motion representation by solving generative tasks (*e.g.* strike, terrain traversal) and motion tracking using VR controllers.

## 1 Introduction

Physically simulated humanoids have a broad range of applications in animation, gaming, AR/VR, robotics, etc., and have seen tremendous improvement in performance and usability in recent years. Due to advances in motion capture (MoCap) and reinforcement learning (RL), humanoids can now imitate entire datasets of human motion (Luo et al., 2023), perform dazzling animations (Peng et al., 2021), and track complex motion from sparse sensors (Winkler et al., 2022). However, each of these tasks requires the careful curation of motion data and training a new physics-based policy from scratch. This process can be time-consuming and engineering heavy, as any issue in reward design, dataset curation, or learning framework can lead to unnatural and jarring motion. Thus, though they can create physically realistic human motion, simulated humanoids remain inapplicable en masse.

One natural way to reuse pre-trained motion controllers is to form a latent space/skill embedding that can be reused in hierarchical RL. First, a task such as motion imitation (Merel et al., 2018; Bohez et al., 2022; Won et al., 2022; Merel et al., 2020; Yao et al., 2022) or adversarial learning (Peng et al., 2022; Tessler et al., 2023; Luo et al., 2020a) is used to form a representation space that can be translated into motor action using a trained decoder. Then, for new tasks, a high-level and task-specific policy is trained to generate latent codes as action, allowing efficient sampling from a structured action space and the reuse of previously learned motor skills. The main issue of this approach is the coverage of the learned latent space. Previous methods use small and specialized motion datasets and focus on specific styles of movements like locomotion, boxing, or dancing (Peng et al., 2022; Tessler et al., 2023; Hasenclever et al.; Zhu et al., 2023; Won et al., 2022; Yao et al., 2022). This specialization yields latent spaces that can only produce a narrow range of behaviors

---

*Equal Advising.

Figure 1: We propose to learn a motion representation that can be reused universally by downstream tasks. From left to right: speed, strike target, complex terrain traversal, and VR controller tracking.

preordained by the training data. Attempts to expand the training dataset, such as to CMU MoCap (CMU, 2002), have not yielded satisfactory results (Won et al., 2022; Yao et al., 2022). Thus, these motion representations can only be applied to generative tasks such as locomotion and stylized animation. For tasks such as free-form motion tracking, they are limited by their motion latent spaces' coverage of the wide spectrum of possible human motion.

Another way to reuse motor skills is to formulate the low-level controller as a motion *imitator*, with the high-level control signal being full-body kinematic motion. This approach is frequently used in motion tracking methods (Yuan et al., 2022; 2021; Won et al., 2020; Zhang et al., 2023a; Luo et al., 2022; 2021; 2023), and trained with supervised learning with paired input (*e.g.* video and kinematic motion). In generative tasks without paired data, RL is required. However, kinematic motion is a poor sampling space for RL due to its high dimension and lack of physical constraints (*e.g.* a small change in root position can lead to a large jump in motion). To apply this approach to generative tasks, an additional kinematic motion latent space such as MVAE (Zhang et al., 2023a; Ling et al., 2020) or HuMoR (Rempe et al., 2021) is needed. Additionally, when agents must interact with their environment or other agents, using only kinematics as a motion signal does not provide insight into interaction dynamics. This makes tasks such as object manipulation and terrain traversal difficult.

In this work, we demonstrate the feasibility of learning a universal motion representation for humanoid control. Our proposed **P**hysics-based **U**niversal motion **L**atent **S**pac**E** (PULSE) is akin to a foundation model for control where downstream tasks ranging from simple locomotion, complex terrain traversal, to free-form motion tracking can all reuse this representation. To achieve this, we leverage recent advances in motion imitation (Luo et al., 2023) and distill comprehensive motor skills into a probabilistic latent space using a variational information bottleneck. This formulation ensures that the learned latent space inherits the motor skills of the motion imitator trained on a large-scale dataset. Our key insight lies in using a pretrained imitator for direct online distillation, which is crucial for effective scaling to large datasets.

To enhance our model's expressiveness, we jointly train a learnable prior conditioned on proprioception (humanoid's own pose and velocities). Random rollouts from this prior using Gaussian noise lead to diverse and stable humanoid motion. This is essential to improve sampling efficiency for downstream tasks since the high-level controller now samples coherent human motion for exploration instead of relying on applying noisy joint torques. The use of this prior in hierarchical RL speeds up training for the motion tracking task and leads to human-like behavior for generative tasks such as complex terrain traversal (Rempe et al., 2023) without using an adversarial reward.

To summarize, our contributions are as follows: 1) We show that one can form a *universal* humanoid motion latent space by distilling from an imitator that can imitate *all* of the motion from a large scale motion dataset. 2) We show that random rollouts from such a latent space can lead to human-like motion. 3) We apply this motion representation to generative tasks ranging from simple locomotion to complex terrain traversal and show that it can speed up training and generate realistic human-like behavior when trained with simplistic rewards. When applied to motion tracking tasks such as VR controller tracking, our latent space can produce free-from human motion.

## 2 RELATED WORK

**Physics-based Humanoid Motion Latent Space**. Recent advances often use adversarial learning or motion tracking objective to form reusable motion representation. Among them, ASE (Peng et al., 2022) and CALM (Tessler et al., 2023) use a discriminator or encoder reward to encourage mapping between random noise and realistic behavior. Although effective on small and specialized datasets (such as locomotion and sword strikes), the formed latent space can fail to cover more diverse motor skills. Using explicit motion tracking, on the other hand, can potentially form a latent space that

covers all the motor skills from the data. ControlVAE (Yao et al., 2022), NPMP (Merel et al., 2018), PhysicsVAE(Won et al., 2022) and NCP (Zhu et al., 2023) all follow this methodology. NCP learns the latent space jointly through the imitation task using RL, while ControlVAE and PhysicsVAE use an additionally learned world model. They show impressive results on tasks like point-goal, maze traversal, boxing, etc, but for tasks such as handling uneven terrains, adaptation layers are required (Won et al., 2022). We also use motion tracking as the task to learn motion representation, but form a *universal* latent space that covers 40 hours of MoCap data. We show that the key to a universal latent space is distilling from a pretrained imitator and a jointly learned prior.

**Kinematics-based Human Motion Latent Space**. Kinematics-based motion latent space has achieved great success in motion generation and pose estimation. In these approaches, motion representation is learned directly from data (without involving physics simulation) via supervised learning. Some methods obtain latent space by compressing multi-step motion into a single code (Li et al., 2021; Wang et al., 2021; Yuan & Kitani, 2020b; Luo et al., 2020b; Petrovich et al., 2021; Guo et al., 2022; Zhang et al., 2023b; Jiang et al., 2023; Lucas et al., 2022). Some, like MVAE (Ling et al., 2020) and HuMoR (Rempe et al., 2021), are autoregressive models that model human motion at the per-frame level and are more applicable to interactive control tasks. While MVAE is primarily geared toward locomotion tasks, HuMoR is trained on AMASS (Mahmood et al., 2019) and can serve as a universal motion prior. Our formulation is close to HuMoR while involving control dynamics. Through random sampling, our latent space governed by the laws of physics can generate long and physically plausible motion, while HuMoR can often lead to implausible ones.

**Physics-based Humanoid Motion tracking**. From DeepMimic (Peng et al., 2018), RL-based motion tracking has gone from imitating single clips to large-scale datasets (Chentanez et al., 2018; Wang et al., 2020; Luo et al., 2023; Fussell et al., 2021). Among them, a mixture of experts (Won et al., 2020), differentiable simulation (Ren et al., 2023), and external forces (Yuan & Kitani, 2020a) have been used to improve the quality of motion imitation. Recently, Perpetual Humanoid Controller (PHC) (Luo et al., 2023) allows a single policy to mimic almost all of AMASS and recover from falls. We improve PHC to track all AMASS and distill its motor skills into a latent space.

**Knowledge Transfer and Policy Distillation**. In Policy Distillation (Rusu et al., 2015), a student policy is trained using teacher's collected experiences via supervised learning (SL). In kickstarting (Schmitt et al., 2018), the student collects its own experiences and is trained with both RL and SL. As we distill motor skills from a pretrained imitator, we also use the student for exploration and the teacher to annotate the collected experiences, similar to DAgger (Ross et al., 2010).

## 3 PRELIMINARIES

We define the full-body human pose as $q_t \triangleq (\theta_t, p_t)$, consisting of 3D joint rotation $\theta_t \in \mathbb{R}^{J \times 6}$ and position $p_t \in \mathbb{R}^{J \times 3}$ of all $J$ links on the humanoid, using the 6 degree-of-freedom (DOF) rotation representation (Zhou et al., 2019). To describe the movement of human motion, we include velocities $\dot{q}_{1:T}$, where $\dot{q}_t \triangleq (\omega_t, v_t)$ consists of angular $\omega_t \in \mathbb{R}^{J \times 3}$ and linear velocities $v_t \in \mathbb{R}^{J \times 3}$. As a notation convention, we use $\hat{\cdot}$ to denote the ground truth kinematic quantities from Motion Capture (MoCap) and normal symbols without accents for values from the physics simulation. A MoCap dataset is called $\hat{Q}$. We use the terms "track", "mimic", and "imitate" interchangeably.

**Goal-conditioned Reinforcement Learning for Humanoid Control**. In this work, all tasks use the general framework of goal-conditioned RL. Namely, a goal-conditioned policy $\pi$ is trained to complete tasks such as imitating the reference motion $\hat{q}_{1:T}$, following 2D trajectories $\tau_{1:T}$, tracking 3D VR controller trajectories $p_{1:T}^{\text{VR}}$, etc. The learning task is formulated as a Markov Decision Process (MDP) defined by the tuple $\mathcal{M} = \langle \mathcal{S}, \mathcal{A}, \mathcal{T}, \mathcal{R}, \gamma \rangle$ of states, actions, transition dynamics, reward function, and discount factor. Physics simulation determines the state $s_t \in \mathcal{S}$ and transition dynamics $\mathcal{T}$, where a policy computes the action $a_t$. For humanoid control tasks, the state $s_t$ contains the proprioception $s_t^{\text{p}}$ and the goal state $s_t^{\text{g}}$. Proprioception is defined as $s_t^{\text{p}} \triangleq (q_t, \dot{q}_t)$, which contains the 3D body pose $q_t$ and velocity $\dot{q}_t$. The goal state $s_t^{\text{g}}$ is defined based on the current task. When computing the states $s_t^{\text{g}}$ and $s_t^{\text{p}}$, all values are normalized with respect to the humanoid heading (yaw). Based on the proprioception $s_t^{\text{p}}$ and the goal state $s_t^{\text{g}}$, we have a reward $r_t = \mathcal{R}(s_t^{\text{p}}, s_t^{\text{g}})$ for training the policy. We use proximal policy optimization (PPO) (Schulman et al., 2017) to maximize discounted reward $\mathbb{E}\left[\sum_{t=1}^{T} \gamma^{t-1} r_t\right]$. Our humanoid follows the kinematic

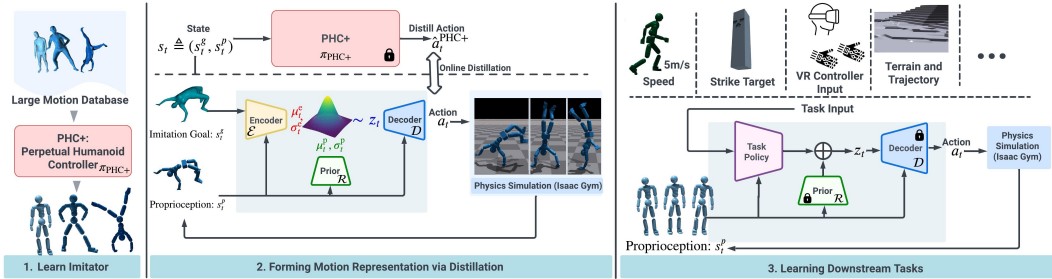

Figure 2: We form our latent space by directly distilling from a pretrained motion imitator that can imitate all of the motion sequences from a large-scale dataset. A variational information bottleneck is used to model the distribution of motor skills conditioned on proprioception. After training the latent space model, the decoder $\mathcal{D}$ and prior $\mathcal{R}$ are frozen and used for downsteam tasks.

structure of SMPL (Loper et al., 2015) using the mean shape. It has 24 joints, of which 23 are actuated, resulting in an action space of $\boldsymbol{a}_t \in \mathbb{R}^{23 \times 3}$. Each degree of freedom is actuated by a proportional derivative (PD) controller, and the action $\boldsymbol{a}_t$ specifies the PD target.

# 4 PHYSICS-BASED UNIVERSAL HUMANOID MOTION LATENT SPACE

To form the universal humanoid motion representation, we first learn to imitate all motion from a large-scale motion dataset (Sec.4.1). Then, we distill the motor skills of this imitator into a latent space model using a variational information bottleneck (4.2). Finally, after training this latent space model, we use it as an action space for downstream tasks (4.3). The pipeline is visualized in Fig.2.

## 4.1 PHC+: ONE POLICY TO IMITATE THEM ALL

For motion imitation, we follow the Perpetual Humanoid Controller (PHC Luo et al. (2023)) and train a motion imitator that can recover from fallen states. $\pi_{\text{PHC}}(\boldsymbol{a}_t | \boldsymbol{s}_t^{\text{p}}, \boldsymbol{s}_t^{\text{g-mimic}})$ is formulated as a per-frame motion imitation task, where at each frame, the goal state $\boldsymbol{s}_t^{\text{g-mimic}}$ is defined as the difference between proprioception and one frame reference pose $\hat{\boldsymbol{q}}_{t+1}$: $\boldsymbol{s}_t^{\text{g-mimic}} \triangleq \hat{\boldsymbol{\theta}}_{t+1} \ominus \boldsymbol{\theta}_t, \hat{\boldsymbol{p}}_{t+1} - \boldsymbol{p}_t, \hat{\boldsymbol{v}}_{t+1} - \boldsymbol{v}_t, \hat{\boldsymbol{\omega}}_t - \boldsymbol{\omega}_t, \hat{\boldsymbol{\theta}}_{t+1}, \hat{\boldsymbol{p}}_{t+1})$. PHC utilizes a progressive training strategy, where a primitive policy $\mathcal{P}^{(0)}$ is first trained on the entire motion dataset $\hat{\boldsymbol{Q}}$. The success rate is regularly evaluated on $\hat{\boldsymbol{Q}}$ and after it plateaus, sequences in which $\mathcal{P}^{(0)}$ fails on form $\hat{\boldsymbol{Q}}_{\text{hard}}^{(0)}$, and a new primitive $\mathcal{P}^{(1)}$ is initialized to learn $\hat{\boldsymbol{Q}}_{\text{hard}}^{(0)}$. This process continues until $\hat{\boldsymbol{Q}}_{\text{hard}}^{(t)}$ cannot be learned or contains no motion. To learn to recover from the fail-state, an additional primitive $\mathcal{P}^{(F)}$ is trained. After learning the primitives, a composer $\mathcal{C}$ is trained to dynamically switch between frozen primitives. PHC achieves a 98.9% success rate on the AMASS training data, and we make small, yet critical modifications to reach 100% to form our imitator, PHC+.

First, extensive analysis of PHC reveals that the MoCap training dataset still contains sequences with severe penetration and discontinuity. These motions doubly impede PHC due to its hard-negative mining procedure, and removing them improves performance. Second, by immediately initializing $\mathcal{P}^{(t+1)}$ to learn from $\hat{\boldsymbol{Q}}_{\text{hard}}^{(t)}$, PHC deprives $\mathcal{P}^{(t)}$ of any opportunity to *further* improve using its own hard negatives. Focusing $\mathcal{P}^{(t)}$ on $\hat{\boldsymbol{Q}}_{\text{hard}}^{(t)}$ before instantiating $\mathcal{P}^{(t+1)}$ better utilizes $\mathcal{P}^{(t)}$'s capacity, and coupled with some modernizing architectural changes, we use only three primitives to learn fail-state recovery and achieve a success rate of 100%. More details are in the supplementary.

## 4.2 LEARNING MOTION REPRESENTATION VIA ONLINE DISTILLATION

After learning PHC+ $\pi_{\text{PHC+}}(\boldsymbol{a}_t^{\text{PHC+}} | \boldsymbol{s}_t^{\text{p}}, \boldsymbol{s}_t^{\text{g-mimic}})$ through RL, we learn a latent representation via on-line distillation. We adopt an encoder-decoder structure and use a conditional variational information bottleneck to model the distribution of motor skills $P(\boldsymbol{a}_t | \boldsymbol{s}_t^{\text{p}}, \boldsymbol{s}_t^{\text{g-mimic}})$ produced in $\pi_{\text{PHC+}}$. Although similar to the VAE architecture, our model does not use any reconstruction loss and is directly optimized in the action $\boldsymbol{a}_t$ space. Specifically, we have a variational encoder $\mathcal{E}(\boldsymbol{z}_t | \boldsymbol{s}_t^{\text{p}}, \boldsymbol{s}_t^{\text{g-mimic}})$ that computes the latent code distribution based on current input states and a decoder $\mathcal{D}(\boldsymbol{a}_t | \boldsymbol{s}_t^{\text{p}}, \boldsymbol{z}_t)$ that produces action. Inspired by HuMoR (Rempe et al., 2021), we employ a learned conditional prior $\mathcal{R}(\boldsymbol{z}_t | \boldsymbol{s}_t^{\text{p}})$ instead of the zero-mean Gaussian prior used in VAEs. The learnable prior $\mathcal{R}$ allows

the model to learn different distributions based on proprioception $s_t^{\mathrm{p}}$, as the action distribution for a person standing still and flipping midair can be significantly different. Formally, we model the encoder and prior distribution as diagonal Gaussian:

$$\boldsymbol{\mathcal{E}}(z_t|s_t^{\mathrm{p}}, s_t^{\mathrm{g\text{-}mimic}}) = \mathcal{N}(z_t|\boldsymbol{\mu}_t^e, \boldsymbol{\sigma}_t^e), \boldsymbol{\mathcal{R}}(z_t|s_t^{\mathrm{p}}) = \mathcal{N}(z_t|\boldsymbol{\mu}_t^p, \boldsymbol{\sigma}_t^p). \tag{1}$$

During training, we sample latent codes from the encoder distribution $z_t \sim \mathcal{N}(z_t|\boldsymbol{\mu}_t^e, \boldsymbol{\sigma}_t^e)$ and use the decoder $\boldsymbol{\mathcal{D}}$ to compute the action. Combining the three networks, we obtain the student policy $\pi_{\mathrm{PULSE}} \triangleq (\boldsymbol{\mathcal{E}}, \boldsymbol{\mathcal{D}}, \boldsymbol{\mathcal{R}})$. Together with the learnable prior, the objective can be written as:

$$\log P(a_t|s_t^{\mathrm{p}}, s_t^{\mathrm{g\text{-}mimic}}) \geq E_{\boldsymbol{\mathcal{E}}}[\log \boldsymbol{\mathcal{D}}(a_t|s_t^{\mathrm{p}}, z_t)] - D_{\mathrm{KL}}(\boldsymbol{\mathcal{E}}(z_t|s_t^{\mathrm{p}}, s_t^{\mathrm{g\text{-}mimic}})||\boldsymbol{\mathcal{R}}(z_t|s_t^{\mathrm{p}})) \tag{2}$$

using the evidence lower bound. The data term $E_{\boldsymbol{\mathcal{E}}}$ is similar to the reconstruction term in VAEs, and the KL term encourages the distribution of the latent code to be close to the learnable prior. To optimize this object, the loss function is written as:

$$\mathcal{L} = \mathcal{L}_{\mathrm{action}} + \alpha\mathcal{L}_{\mathrm{regu}} + \beta\mathcal{L}_{\mathrm{KL}}, \tag{3}$$

which contains the data term $\mathcal{L}_{\mathrm{action}}$, the learnable prior $\mathcal{L}_{\mathrm{KL}}$ from Eq.2, and a regulization term $\mathcal{L}_{\mathrm{regu}}$. For online distillation, we have $\mathcal{L}_{\mathrm{action}} = \|a_t^{\mathrm{PHC+}} - a_t\|_2^2$, similar to the reconstruction term in VAEs. In our experiments, we find that if unconstrained, the learnable prior can push the latent codes far away from each other even for $s_t^{\mathrm{p}}$ that are close. Thus, we introduce an additional regularization term $\mathcal{L}_{\mathrm{regu}} = \|\boldsymbol{\mu}_t^e - \boldsymbol{\mu}_{t-1}^e\|_2^2$ that penalizes a large deviation of consecutive latent codes. Intuitively, temporally close transitions should have similar latent representations, providing a smoother and more continuous latent space. $\mathcal{L}_{\mathrm{regu}}$ serves as a weak prior, similar to the AR(1) prior used in NPMP (Merel et al., 2018). In ablations, we show that using this prior is crucial for downstream tasks, without which the latent space can be very discontinuous and hard to navigate during RL exploration. To maintain the delicate balance of reconstruction error and KLD, we anneal $\beta$ gradually.

For training $\pi_{\mathrm{PULSE}}$, we need pairs of $(s_t^{\mathrm{p}}, s_t^{\mathrm{g\text{-}mimic}}, a_t, a_t^{\mathrm{PHC+}})$, which are obtained by rolling out $\pi_{\mathrm{PULSE}}$ in the environment for the motion imitation task and querying $\pi_{\mathrm{PHC+}}$. While we can optimize $\pi_{\mathrm{PULSE}}$ with the RL objective by treating the decoder $\boldsymbol{\mathcal{D}}$ as a Gaussian distribution with fixed diagonal covariance matrix (similar to kickstarting (Schmitt et al., 2018)), we find that sampling the latent code together with random exploration sampling for RL introduces too much instability to the system. In our experiments, we show that training with RL objectives alone is not sufficient to form a good latent space, and optimizing with RL and supervised objectives combined creates a noisy latent space that is worse than without. The same as PHC+'s training process, we perform hard-negative mining and form $\hat{Q}_{\mathrm{hard}}$ for progressive training. When evaluating $\pi_{\mathrm{PULSE}}$, we use the mean latent code $\boldsymbol{\mu}_t^e$ instead of sampling from $\mathcal{N}(z_t|\boldsymbol{\mu}_t^e, \boldsymbol{\sigma}_t^e)$. This progressive training adds the benefit of balancing the data samples of the rarer and harder training motion sequences with the simpler ones.

### 4.3 HIERARCHICAL CONTROL FOR DOWNSTREAM TASKS

After $\pi_{\mathrm{PULSE}}$ has converged, we can train a high-level policy $\pi_{\mathrm{task}}(z_t^{\mathrm{task}}|s_t^{\mathrm{p}}, s_t^{\mathrm{g}})$ for downstream tasks. The frozen decoder $\boldsymbol{\mathcal{D}}$, together with the simulation, can now be treated as the new dynamcis system (Haarnoja et al., 2018), where the action is now in the $z_t$ space. Each high-level task policy $\pi_{\mathrm{task}}(z_t^{\mathrm{task}}|s_t^{\mathrm{p}}, s_t^{\mathrm{g}}) = \mathcal{N}(\boldsymbol{\mu}^{\mathrm{task}}(s_t^{\mathrm{p}}, s_t^{\mathrm{g}}), \boldsymbol{\sigma}^{\mathrm{task}})$ represents a Gaussian distribution with fixed diagonal covariance. We test on a suite of generative and motion tasks, and show that sampling from our prior, our policy can create realistic human-like behavior for downstream tasks, without relying on any adversarial learning objectives. The full training pipeline can be found in Alg.1.

**Hierarchical Control with Learnable Prior**. During high-level policy $\pi_{\mathrm{task}}$ training, it is essential to sample action from the prior distribution induced by $\boldsymbol{\mathcal{R}}$. Intuitively, if PULSE's latent space encapsulates a broad range of motor skills, randomly sampled codes may not lead to coherent motion. The learnable prior provides a good starting point for sampling human-like behavior. To achieve this, we form the action space of $\pi_{\mathrm{task}}$ as the residual action with respect to prior's mean $\boldsymbol{\mu}_t^p$ and compute the PD target $a_t^{\mathrm{task}}$ for downstream tasks as:

$$a_t^{\mathrm{task}} = \boldsymbol{\mathcal{D}}(\pi_{\mathrm{task}}(z_t^{\mathrm{task}}|s_t^{\mathrm{p}}, s_t^{\mathrm{g}}) + \boldsymbol{\mu}_t^p), \tag{4}$$

where $\boldsymbol{\mu}_t^p$ is computed by the prior $\boldsymbol{\mathcal{R}}(z_t|s_t^{\mathrm{p}})$. For RL exploration for $\pi_{\mathrm{task}}$, we use a fixed variance (0.22) instead of $\boldsymbol{\sigma}_t^p$ as we notice that $\boldsymbol{\sigma}_t^p$ tends to be rather small. In ablations, we show that not using this residual action formulation for downstream task leads to a significant drop in performance.

---

**Algo 1:** Learn PULSE and Downstream Tasks

---

1 **Function** `TrainPULSE`($\mathcal{E}, \mathcal{D}, \mathcal{R}, \hat{Q}, \pi_{PHC+}$)**:**
2     **Input:** Ground truth motion dataset $\hat{Q}$, pretrained PHC+ $\pi_{\text{PHC+}}$, encoder $\mathcal{E}$, decoder $\mathcal{D}$, and prior $\mathcal{R}$ ;
3     **while** *not converged* **do**
4         $M \leftarrow \emptyset$ initialize sampling memory ;
5         **while** $M$ *not full* **do**
6             $\hat{q}_{1:T}, s_t^{\text{p}} \leftarrow$ sample motion and initial state from $\hat{Q}$ ;
7             **for** $t \leftarrow 1 \dots T$ **do**
8                 $s_t \leftarrow \left(s_t^{\text{p}}, s_t^{\text{g-mimic}}\right)$ ;
9                 $\mu_t^e, \sigma_t^e \leftarrow \mathcal{E}(z_t|s_t^{\text{p}}, s_t^{\text{g-mimic}})$         // encode latent;
10                $z_t \sim \mathcal{N}(z_t|\mu_t^e, \sigma_t^e)$    // reparameterization trick for sampling latents;
11                $a_t \leftarrow \mathcal{D}(a_t|s_t^{\text{p}}, z_t)$       // decode action;
12                $s_{t+1} \leftarrow \mathcal{T}(s_{t+1}|s_t, a_t)$       // simulation;
13             store $s_t$ into memory $M$ ;
14         $a_t^{\text{PHC+}} \leftarrow \pi_{\text{PHC+}}(a_t^{\text{PHC+}}|s_t)$ Annotate collected states in $M$ using $\pi_{\text{PHC+}}$ ;
15         $\mu_t^p, \sigma_t^p \leftarrow \mathcal{R}(z_t|s_t^{\text{p}})$ Compute prior distribution based on proprioception ;
16         $\mathcal{R}, \mathcal{D}, \mathcal{E} \leftarrow$ supervised update for encoder, decoder, and prior using pairs of $(a_t, a_t^{\text{PHC+}}, \mu_t^p, \sigma_t^p, \mu_t^e, \sigma_t^e)$ and Eq.3.
17     **return** $\mathcal{E}, \mathcal{D}, \mathcal{R}$ ;

18 **Function** `TrainDownstreamTask`($\mathcal{D}, \mathcal{R}, \hat{Q}, \pi_{task}$)**:**
19     **Input:** Pretrained PULSE's decoder $\mathcal{D}$ and prior $\mathcal{R}$, task definition, and motion dataset for initial state sampling (*e.g.* $\hat{Q}$) ;
20     **while** *not converged* **do**
21         $M \leftarrow \emptyset$ initialize sampling memory ;
22         **while** $M$ *not full* **do**
23             $s_t^{\text{p}} \leftarrow$ sample initial state from $\hat{Q}$ ;
24             **for** $t \leftarrow 1 \dots T$ **do**
25                 $z_t^{\text{task}} \sim \pi_{\text{task}}(a_t|s_t^{\text{p}}, s_t^{\text{g-task}})$    // usee pretrained latent space as action space;
26                 $\mu_t^p, \sigma_t^p \leftarrow \mathcal{R}(z_t|s_t^{\text{p}})$         // compute prior latent code;
27                 $a_t^{\text{task}} \leftarrow \mathcal{D}(a_t|s_t^{\text{p}}, z_t^{\text{task}} + \mu_t^p)$    // decode action using pretrained decoder;
28                 $s_{t+1} \leftarrow \mathcal{T}(s_{t+1}|s_t, a_t^{\text{task}})$        // simulation;
29                 $r_t \leftarrow \mathcal{R}(s_t, \hat{q}_{t+1})$         // compute reward;
30              store $(s_t, z_t, r_t, s_{t+1})$ into memory $M$ ;
31         $\pi_{\text{task}} \leftarrow$ PPO update using experiences collected in $M$ ;
32     **return** $\pi_{\text{task}}$ ;

---

## 5 EXPERIMENTS

**Tasks**. We study a suite of popular downstream tasks used in the prior art. For generative tasks, we study reaching a certain x-directional speed (between 0 m/s and 5m/s), striking a target with the right hand (target initialized between 1.5∼5m), reaching a 3D point with the right hand (point initialized within 1 meters of the humanoid), and following a trajectory on complex terrains (slope, stairs, uneven surfaces, and obstacles). For motion tracking, we study a VR controller tracking task, where the humanoid tracks three 6DOF rigidbodies in space (two hand controllers and a headset). Note that the VR controller tracking task can be viewed as a generalization of the full-body motion imitation task, where only partial observation is provided. It is challenging since it requires the latent space to contain motor skills to match arbitrary user input in a continuous and streaming fashion. Please refer to the supplement for a complete definition of the tasks.

**Datasets**. For training PHC+, PULSE, and VR controller policy, we use the cleaned AMASS training set. For strike and reach tasks, we use the AMASS training set for initial state sampling. For speed and terrain traversal tasks, we follow Pacer (Rempe et al., 2023) to use a subset of AMASS that contains only locomotion for initial state sampling. All methods are trained using the same initial-state sampling strategy. For testing imitation and VR controller tracking policy, we use the AMASS test set, as well as a real-world dataset (14 sequences, 16 minutes) from QuestSim (Winkler et al., 2022) collected using the Quest 2 headset.

**Baselines**. To compare with state-of-the-art latent space models, we train ASE (Peng et al., 2022) and CALM (Tessler et al., 2023) with the officially released code. We train all the latent space models from scratch using the AMASS training set with around 1 billion samples. Then, we apply the learned latent space to downstream tasks. Both CALM and ASE use a 64-d latent code projected onto a unit sphere, and are designed to be multi-frame latent codes operating at 6 HZ. Since our policy produces a per-frame latent code (30 Hz), for a fair comparison, we also test against ASE and CALM operating at 30 Hz. Notice that by design, CALM is trained to perform downstream tasks

Table 1: Motion imitation result (*data cleaning) on AMASS train and test (11313 and 138 sequences).

| Method | AMASS-Train* | | | | | AMASS-Test* | | | | |
|---|---|---|---|---|---|---|---|---|---|---|
| | Succ ↑ | $E_{\text{g-mpjpe}}$ ↓ | $E_{\text{mpjpe}}$ ↓ | $E_{\text{acc}}$ ↓ | $E_{\text{vel}}$ ↓ | Succ ↑ | $E_{\text{g-mpjpe}}$ ↓ | $E_{\text{mpjpe}}$ ↓ | $E_{\text{acc}}$ ↓ | $E_{\text{vel}}$ ↓ |
| PHC | 98.9 % | 37.5 | 26.9 | 3.3 | 4.9 | 97.1% | 47.5 | 40.0 | 6.8 | 9.1 |
| PHC+ | **100 %** | **26.6** | **21.1** | **2.7** | **3.9** | **99.2%** | **36.1** | **24.1** | **6.2** | **8.1** |
| PULSE | 99.8 % | 39.2 | 35.0 | 3.1 | 5.2 | 97.1% | 54.1 | 43.5 | 7.0 | 10.3 |

(a) Reach   (b) Strike   (c) Speed

(d) Terrain

(e) VR Controller Tracking

(f) A Random Rollout

Figure 3: (a, b, c, d) Policy trained using our motion representation solves tasks with human-like behavior. (e) Our latent space is not constrained to certain movement styles and support free-form tracking. (f) Random sampling from learned prior $\mathcal{R}$ leads to human-like movements as well as the recovery from fallen state.

with user-specified skills. However, since we train our generative task without a specific style in mind, we opt out of the style reward and only train CALM with the task reward. For all tasks, we also compare against a policy trained from scratch without latent space or adversarial reward.

**Metrics**. For generative tasks (reach, speed, strike, and terrain), we show the training curve by visualizing the undiscounted return normalized by the maximum possible reward per episode. For motion imitation and VR controller tracking, we report the popular full body mean per joint position error for both global $E_{\text{g-mpjpe}}$ and root-relative $E_{\text{mpjpe}}$ (in mm), and physics-based metrics such as acceleration error $E_{\text{acc}}$ (mm/frame$^2$) and velocity error $E_{\text{vel}}$ (mm/frame). Following PHC, we measure the success rate Succ (the success rate is defined as following reference motion $< 0.5$ m, averaged per joint, at all points in time). For VR controller tracking, the success rate is measured against only the three body points, but we report full-body $E_{\text{g-mpjpe}}$ and $E_{\text{mpjpe}}$ when they are available (on AMASS). On the real-world dataset without paired full-body motion, we report controller tracking metrics $E_{\text{g-mhpe}}$ (mm) which computes the global position error of the three tracked controllers.

**Implementation Details**. Simulation is conducted at Isaac Gym (Makoviychuk et al., 2021), where the policy is run at 30 Hz and the simulation at 60 Hz. For PHC+ and all downstream tasks, each policy or primitive is a 6-layer MLP. Our encoder $\mathcal{E}$, decoder $\mathcal{D}$, prior $\mathcal{R}$, CALM, and ASE are three-layer MLPs. The latent space is 32d: $z_t \in \mathbb{R}^{32}$, about half of the humanoid DOF of 69.

## 5.1 MOTION IMITATION

First, we measure the imitation quality of PHC+ and PULSE after distilling from PHC+. Table 1 shows that after distillation using the variational bottleneck, PULSE retains most of the motor skills generated by PHC+. This is crucial as this shows that the learned latent space model (the decoder $\mathcal{D}$) inherits most of the motor skills to perform the majority of the AMASS dataset with high accuracy. Note that without the variational bottleneck, distillation *can reach* a 100% success rate, and the degradation in performance is similar to a non-zero reconstruction error of a VAE.

## 5.2 DOWNSTREAM TASKS

**Generative Tasks**. We show the qualitative result in Fig.3 for the generative tasks (speed, reach,

Table 2: Quantitative results on VR Controller tracking. We report result on AMASS and real-world dataset.

| | AMASS-Train* | | | | | AMASS-Test* | | | | | Real-world | | | |
|---|---|---|---|---|---|---|---|---|---|---|---|---|---|---|
| Method | Succ ↑ | $E_{\text{g-mpjpe}}$ ↓ | $E_{\text{mpjpe}}$ ↓ | $E_{\text{acc}}$ ↓ | $E_{\text{vel}}$ ↓ | Succ ↑ | $E_{\text{g-mpjpe}}$ ↓ | $E_{\text{mpjpe}}$ ↓ | $E_{\text{acc}}$ ↓ | $E_{\text{vel}}$ ↓ | Succ ↑ | $E_{\text{g-mhpe}}$ ↓ | $E_{\text{acc}}$ ↓ | $E_{\text{vel}}$ ↓ |
| ASE-30Hz | 79.8% | 103.0 | 80.8 | 12.2 | 13.6 | 37.6% | 120.5 | 83.0 | 19.5 | 20.9 | 7/14 | 99.0 | 28.2 | 20.3 |
| ASE-6Hz | 74.2% | 113.9 | 84.9 | 81.9 | 57.7 | 33.3% | 136.4 | 98.7 | 117.7 | 80.5 | 4/14 | 114.7 | 115.8 | 72.9 |
| Calm-30Hz | 16.6% | 130.7 | 100.8 | **2.7** | **7.9** | 10.1% | 122.4 | 99.4 | **7.2** | **12.9** | 2/14 | 206.9 | **2.0** | **6.6** |
| Calm-6Hz | 14.6% | 137.6 | 103.2 | 58.9 | 47.7 | 10.9% | 147.6 | 115.9 | 82.5 | 122.3 | 0/14 | - | - | - |
| Scratch | 98.8% | **51.7** | **47.9** | 3.4 | 6.4 | **93.4%** | **80.4** | 67.4 | 8.0 | 13.3 | **14/14** | **43.3** | 4.3 | 6.5 |
| Ours | **99.5%** | 57.8 | 51.0 | 3.9 | 7.1 | 93.4 % | 88.6 | 67.7 | 9.1 | 14.9 | **14/14** | 68.4 | 5.8 | 9.0 |

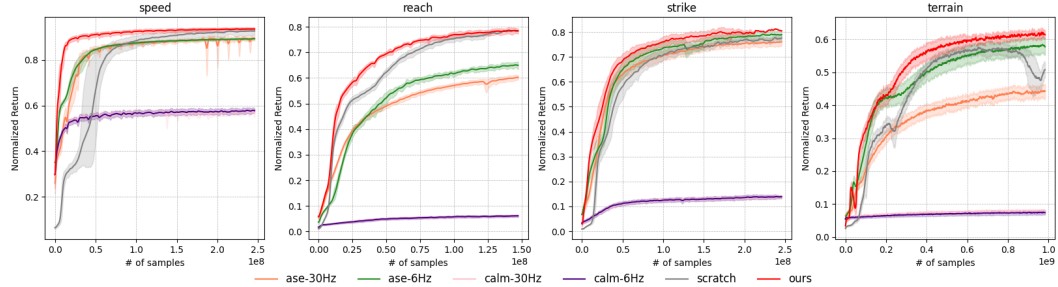

Figure 4: Training curves for each one of the generative tasks. Using our motion representation improves training speed and performance as the task policy can explore in an informative latent space. Experiments run for 3 times using different random seeds.

strike, and terrain). From Fig.4 we can see that using our latent space model as the low-level controller outperforms all baselines, including the two latent space models and training from scratch.

Compared to latent space models (ASE-30Hz, ASE-6Hz, CALM-30Hz, CALM-6Hz), our method consistently provides a better action space, achieving better normalized return across the board. This shows that the ASE and CALM formulation, though effective in learning motion styles from curated datasets, could not adequately capture motor skills in a large and unorganized dataset like AMASS. In general, 30 Hz models (ASE-30Hz and CALM-30Hz) outperform 6 Hz ones, signaling the need for finer-grained control latent space. In general, ASE outperforms CALM, partially due to CALM's motion encoder formulation. Without the style reward, exploration might be hindered for downstream tasks. For all tasks, training from scratch provides the closest normalized return to ours, but this could lead to unnatural behavior, as observed in ASE and our qualitative results. Our method also converges faster, even for

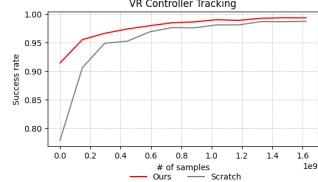

Figure 5: Success rate comparison between training from scratch and using our motion representation during training. Ours converges faster.

the challenging terrain traversal task, where the humanoid needs to handle obstacles and stairs that it has not seen during training. Upon inspection, we can see that the humanoid uses human-like behavior for navigating the obstacles without using any style/adversarial reward as guidance. When stepping on stairs quickly, we can see that our humanoid employs motor skills similar to jumping. For detailed qualitative evaluation, see supplement site.

**VR Controller Tracking**. Unlike generative tasks, we can directly measure the success rate and joint position error for this task. In Table 2, we report the motion tracking result in the training, test, and real-world dataset. Continuing the trend of generative tasks, compared to other latent space models, ours significantly improves tracking error. This validates the hypothesis that PULSE can capture a wide range of human motor skills and can handle free-form motion tracking and pose estimation tasks. On the real-world dataset, our method can faithfully track all sequences. Note that our tracking result is comparable to training from scratch, which signifies that our latent space is diverse enough that it does not handicap exploration. Since the three-point tracking problem is relatively close to the full-body tracking problem, we expect training from scratch to perform better since it is a specialized tracking policy. Using our latent space also seems to be trading success rate for precise tracking, sometimes sacrificing precision in favor of staying stable. Figure 5 shows the training success rate between training from scratch and using our latent space. We can see that, using our latent space, the policy converges faster.

Table 3: Ablation on various strategies of learning the motion representation. We use the challenging VR controller tracking task to demonstrate the applicability of the latent space for downstream tasks. Prior: whether to use a learnable prior. Prior-action: whether to use the residual action with respect to the learned prior $\mathcal{R}$. $\mathcal{L}_{\text{regu}}$: whether to apply $\mathcal{L}_{\text{regu}}$. No RL: whether to train PULSE together with the RL objective.

| | | | | | AMASS-Test | | | | |
| idx | Prior | Prior-action | $\mathcal{L}_{\text{regu}}$ | No RL | Succ $\uparrow$ | $E_{\text{g-mpjpe}} \downarrow$ | $E_{\text{mpjpe}} \downarrow$ | $E_{\text{acc}} \downarrow$ | $E_{\text{vel}} \downarrow$ |
|---|---|---|---|---|---|---|---|---|---|
| 1 | ✗ | ✗ | ✗ | ✓ | 36.9% | 114.6 | 79.4 | 8.1 | 15.8 |
| 2 | ✗ | ✗ | ✓ | ✓ | 45.6% | 115.0 | 77.2 | **7.5** | 15.0 |
| 3 | ✓ | ✓ | ✗ | ✓ | 60.8% | 106.8 | 72.4 | 9.4 | 16.0 |
| 4 | ✓ | ✓ | ✓ | ✗ | 71.0% | 95.1 | 72.8 | 10.4 | 16.6 |
| 5 | ✓ | ✗ | ✓ | ✓ | 18.1% | 108.6 | 83.8 | 11.2 | 16.4 |
| 6 | ✓ | ✓ | ✓ | ✓ | **93.4%** | **88.6** | **67.7** | 9.1 | **14.9** |

## 5.3 RANDOM MOTION GENERATION

As motion is best seen in videos, we show extensive qualitative random sampling and baseline comparison result in our `supplement site` and Fig.3 also shows a qualitative sample. Comparing with the kinematics-based, HuMoR, our method can generate realistic human motion without suffering from frequent implausible motion generation. Compared to other physics-based latent space, ASE and CALM, our representation has more coverage and can generate more natural and realistic motion. By varying the variance of the input noise, we can control the generation process and bias it toward a smoother or more energetic motion.

## 5.4 ABLATIONS

In this section, we study the effects of different components of our framework using the challenging VR controller tracking task. During our experiments, we noticed that while the generative downstream tasks are affected less by the expressiveness of the latent space, the VR controller tracking task is. We train our latent space using the same teacher (PHC+) and train the VR controller tracking policy on the frozen decoder. When comparing Row 1 (R1) vs. R2 as well as R3 vs. R6, we can see that the regularization term $\mathcal{L}_{\text{regu}}$ that penalizes a large deviation between consecutive latent codes is effective in providing a more compact latent space for downstream exploration. Without such regularization, the encoder can create a discontinuous latent space that is harder to explore. R6 trains a policy that does not use the learnable prior but uses the zero-mean Gaussian $\mathcal{N}(0, 1)$ as in VAEs. R2 vs. R6 shows the importance of having a learnable prior, without which downstream sampling could be difficult. R5 vs. R6 shows that using the learnable prior as a residual action described in Eq.4 is essential for exploration, without which the latent space is too hard to sample from (similar to the issues with CALM without style reward). R4 vs. R6 shows that mixing the RL training and objective and online supervised distillation has a negative effect.

## 6 DISCUSSION AND FUTURE WORK

**Failure Cases**. While PULSE demonstrated that it is feasible to learn a universal latent space model, it is far from perfect. First, $\pi_{\text{PULSE}}$ does not reach a 100% imitation success rate, meaning that the information bottleneck is a lossy compression of the motor skill generated by PHC+. In our experiments, online distillation can reach 100% success rate using a unconstrained latent space (without the variational information bottleneck). The variational formulation enables probabilistic modeling of motor skills, but introduces additional challenges. For downstream tasks, while our method can lead to human-like behavior, for tasks like VR controller tracking, it can lag behind the performance of training from scratch. For motion generation, the humanoid can be stuck in a fallen or standing position, although increasing the noise into the system could jolt it out of these states.

**Future Work**. In conclusion, we present PULSE, a universal motion representation for physics-based humanoid control. It can serve as a generative motion prior for downstream generative and estimation tasks alike, without the use of any additional modification or adaptive layers. Future work includes learning a more human-interpretable motion representation, incorporating scene information and human-object interactions, as well as articulated fingers.

**Acknowledgement**. We thank Zihui Lin for her help in making the plots in this paper. Zhengyi Luo is supported by the Meta AI Mentorship (AIM) program.

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

## APPENDICES

## A  INTRODUCTION

In this document, we include additional details about our method that are omitted from the main paper due to the page limit. In Sec.B, we include additional details about PHC+ and our modifications made to imitate all motion from a large-scale dataset. In Sec.C, we include additional details about our method, PULSE, such as architecture, training details, and downstream task configurations *etc*.**All code and models will be released for research purposes.**

Extensive qualitative results are provided on the project page as well as in the supplementary zip (the zipped version is of lower video resolution to fit the upload size). As motion is best seen in videos, we highly encourage the readers to view them to better understand the capabilities of our method. Specifically, we evaluate motion imitation and fail-state recovery capabilities for PHC+ and PULSE after online distillation and show that PULSE can largely retain the abilities of PHC+. Then, we show long-formed motion generation result sampling from the PULSE' s prior and decoder. Sampling from PULSE, we can generate long-term, diverse and human-like motions, and we can vary the variance of the input noise to control the behaviors of random generation. We also compare with SOTA kinematics-based method, HuMoR, and show that while HuMoR can generate unnatural motions, ours are regulated by the laws of physics and remain plausible. Compared to SOTA physics-based latent space(ASE and CALM), our random generation appears more diverse. Finally, we show visualization for downstream tasks for both generative and estimation/tracking tasks and compare them with SOTA methods.

## B  DETAILS ABOUT PHC+

### B.1  DATA CLEANING

We perform a failure case analysis and identified two main sources of imitation failure. First, we have dynamic motion, such as cartwheeling and consecutive back flips. Another, often overlooked, is that MoCap sequences can still have a large discontinuity and penetration due to failures in the MoCap optimization procedure or the fitting process (Loper et al., 2014). After filtering out human-object interaction data following UHC (Luo et al., 2021), we found additional corrupted sequences in PHC's training data that have a large discontinuity or penetration. In Fig.6, we visualize some of the sequences we have identified and removed from the training data. Since we use the random state initialization proposed by DeepMimic (Peng et al., 2018), sampling frames that have large penetration could lead to the humanoid "flying off" from the ground as the physics simulation applies a large ground reactionary force. Naively adjusting the height of the sequence based on penetration could lead to floating sequences or discontinuity. Frames that have large discontinuities could lead

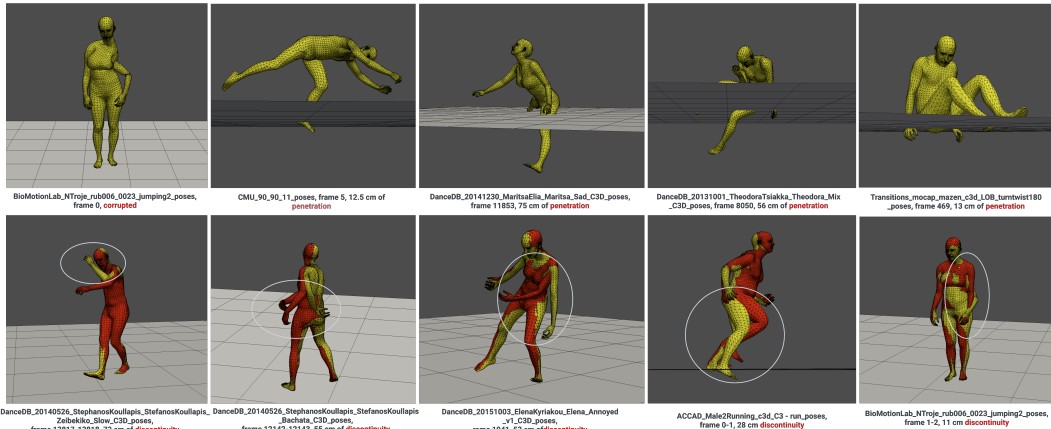

Figure 6: Visualization of issues in the AMASS dataset. Here we show sequences with corrupted poses, large penetration, and discontinuity. In the second row, the red and yellow mesh are 1 frame apart in 120Hz MoCap.

Table 4: Hyperparameters for PHC+ and Pulse. $\sigma$: fixed variance for policy. $\gamma$: discount factor. $\epsilon$: clip range for PPO. $\alpha$: coefficient for $\mathcal{L}_{\text{regu}}$. $\beta$: coefficient for $\mathcal{L}_{\text{KL}}$.

| Method | Batch Size | Learning Rate | $\sigma$ | $\gamma$ | $\epsilon$ | $w_{\text{jp}}$ | $w_{\text{jr}}$ | $w_{\text{jv}}$ | $w_{\text{j}\omega}$ | # of samples |
|--------|-----------|---------------|----------|----------|------------|-----------------|-----------------|-----------------|----------------------|--------------|
| PHC+ | 3072 | $2 \times 10^{-5}$ | 0.05 | 0.99 | 0.2 | 0.5 | 0.3 | 0.1 | 0.1 | $\sim 10^{10}$ |
| | Batch Size | Learning Rate | $\alpha$ | $\beta$ | Latent size | # of samples | | | | |
| PULSE | 3072 | $5 \times 10^{-4}$ | 0.005 | $0.01 \rightarrow 0.001$ | 32 | $\sim 10^9$ | | | | |

to imitation failure or humanoid learning bad behavior to anticipate large jumps between frames. We remove these sequences and obtain 11313 training and 138 testing motion suitable for motion imitation training and testing, which we will release with the code and models.

## B.2 ACTION AND REWARDS

Our action space, state, and rewards follow the specifications of the PHC paper. Specifically, the action $\boldsymbol{a}_t$ specifies the target for the proportional derivative (PD) controller on each of the 69 actuators. The target joint is set to $\boldsymbol{q}_t^d = \boldsymbol{a}_t$ and the torque applied at each joint is $\boldsymbol{\tau}^i = \boldsymbol{k}^p \circ (\boldsymbol{a}_t - \boldsymbol{q}_t) - \boldsymbol{k}^d \circ \dot{\boldsymbol{q}}_t$. Joint torques are capped at 500 N-m. For the motion tracking reward, we use:

$$r_t = 0.5 r_t^{\text{g-imitation}} + 0.5 r_t^{\text{amp}} + r_t^{\text{energy}},$$
$$r_t^{\text{g-imitation}} = w_{\text{jp}} e^{-100\|\hat{\boldsymbol{p}}_t - \boldsymbol{p}_t\|} + w_{\text{jr}} e^{-10\|\hat{\boldsymbol{q}}_t \ominus \boldsymbol{q}_t\|} + w_{\text{jv}} e^{-0.1\|\hat{\boldsymbol{v}}_t - \boldsymbol{v}_t\|} + w_{\text{j}\omega} e^{-0.1\|\hat{\boldsymbol{\omega}}_t - \boldsymbol{\omega}_t\|}, \tag{5}$$

where $r_t^{\text{g-imitation}}$ is the motion imitation reward, $r_t^{\text{amp}}$ is the discriminator reward, and $r_t^{\text{energy}}$ an energy penalty. $r_t^{\text{g-imitation}}$ measures the difference between the translation, rotation, linear velocity, and angular velocity of the 23 rigid bodies in the humanoid. $r_t^{\text{amp}}$. $r_t^{\text{amp}}$ is the Adversarial Motion Prior (AMP) reward (Peng et al., 2021), provided by a discriminator trained on the AMASS dataset. The energy penalty $r_t^{\text{energy}}$ is $-0.0005 \cdot \sum_{j \in \text{joints}} |\boldsymbol{\mu}_j \boldsymbol{\omega}_j|^2$ where $\boldsymbol{\mu}_j$ and $\boldsymbol{\omega}_j$ correspond to the joint torque and the joint angular velocity, respectively. The energy penalty Fu et al. (2022) regulates the policy and prevents high-frequency jitter.

## B.3 MODEL ARCHITECTURE AND ABLATIONS

All primitives and composers in PHC + are a 6 layer MLP with units [2048, 1536, 1024, 1024, 512, 512] and SiLU activation. We find that changing the activation from ReLU (Fukushima, 1975; Nair & Hinton, 2010) to SiLU (Hendrycks & Gimpel, 2016) provides a non-trivial boost in tracking performance. Combining with larger networks (from 3 layer MLP to 6 layer), we use only three

Table 6: Ablations on training PULSE from scratch using RL (no distillation).

| | AMASS-Train* | | | | AMASS-Test* | | | | |
|---|---|---|---|---|---|---|---|---|---|
| Distill | Succ ↑ | $E_{\text{g-mpjpe}} \downarrow$ | $E_{\text{mpjpe}} \downarrow$ | $E_{\text{acc}} \downarrow$ | $E_{\text{vel}} \downarrow$ | Succ ↑ | $E_{\text{g-mpjpe}} \downarrow$ | $E_{\text{mpjpe}} \downarrow$ | $E_{\text{acc}} \downarrow$ | $E_{\text{vel}} \downarrow$ |
| ✗ | 72.0% | 76.7 | 52.8 | 3.5 | 8.0 | 32.6% | 98.4 | 79.4 | 9.9 | 16.2 |
| ✓ | 99.8 % | 39.2 | 35.0 | 3.1 | 5.2 | 97.1% | 54.1 | 43.5 | 7.0 | 10.3 |

primitives to learn fail-state recovery and achieve a success rate of 100%. To study the effect of the new activation function and the progressive training procedure, we perform ablation studies on the training of a single primitive $\mathcal{P}$ (not the full PHC + policy) using the proposed changes.

Each primitive is trained for $3 \times 10^9$ samples. From Table 5, we can see that comparing Row (R1) and R3, the new progressive training procedure improves the success rate by a large amount, showing that $\mathcal{P}$'s capacity is not fully utilized if $\hat{Q}_{\text{hard}}$ is not formed and updated during each $\mathcal{P}$'s training. When comparing R2 and R3, we can see that changing the activation function from ReLU to SiLU improves the tracking performance and improves $E_{\text{mpjpe}}$. Table.4 reports the hyperparameters we used for training.

Table 5: Ablations on PHC+'s primitive $\mathcal{P}$ training. Progressive: refers to whether $\hat{Q}_{\text{hard}}$ is updated during the primitive training (rather than waiting until convergence and initialize a new primitive).

| | | AMASS-Test | | | | |
|---|---|---|---|---|---|---|
| Activation | Progressive | Succ ↑ | $E_{\text{g-mpjpe}} \downarrow$ | $E_{\text{mpjpe}} \downarrow$ | $E_{\text{acc}} \downarrow$ | $E_{\text{vel}} \downarrow$ |
| SiLU | ✗ | 92.0% | 43.0 | 29.2 | 6.7 | 8.9 |
| ReLU | ✓ | 97.8% | 44.4 | 32.8 | 6.9 | 9.1 |
| SiLU | ✓ | **98.5%** | **39.0** | **28.1** | **6.7** | **8.5** |

## C  DETAILS ABOUT PULSE

### C.1  TRAINING PROCEDURE

We train $\pi_{\text{PULSE}}$ using the training procedure we used to train a primitive $\mathcal{P}^{(0)}$ in PHC+, where we progressively form $\hat{Q}_{\text{hard}}$ while training the policy. Since $\pi_{\text{PULSE}}$ and $\pi_{\text{PHC+}}$ share the same state and action space, we query $\pi_{\text{PHC+}}$ at training time to perform online distillation. We anneal the coefficient $\beta$ of $\mathcal{L}_{\text{KL}}$ from 0.01 to 0.001 starting from $2.5 \times 10^9$ to $5 \times 10^9$ samples. Afterward, $\beta$ remains the same. We report our hyperparameters for training $\pi_{\text{PULSE}}$ in Table. 4.

### C.2  COMPARISON TO TRAINING SCRATCH WITHOUT DISTILLATION

One of our main contributions for PULSE is the using online distillation to learn $\pi_{\text{PULSE}}$, where the latent space uses knowledge distilled from a trained imitator, PHC+. While prior work like MCP (Peng et al., 2019) demonstrated the possibility of training such a policy from scratch (using RL without distillation), we find that using the variational information bottleneck together with the imitation objective creates instability during training. We hypothesize that random sampling for the variational bottleneck together with random sampling for RL leads to noisy gradients. In Table 6, we report the result of motion imitation from training from scratch. We can see that the training using RL does not converge to a good imitation policy after training for more than $1 \times 10^{10}$ samples.

### C.3  COMPARISON TO OTHER LATENT FORMULATION (VQ-VAE, SPHERICAL)

In our earlier experiments, we studied other forms of latent space such as a spherical latent space, similar to ASE (Peng et al., 2022), or a vector quantized latent space, similar to NCP (**?**). For spherical embedding, we use the same encoder-decoder structure as in PULSE and use a 32-dimensional latent space normalized to the unit sphere. For a vector-quantized motion representation, we follow Liu et al. (2021); Van Den Oord et al. (2017) and use a 64-dimensional latent space divided into 8 partitions, using a dictionary size of 64. Dividing the latent space into partitions increases the representation power combinatorially (Liu et al., 2021) and is more effective than a larger dictionary size. Although through distillation, each of these representations could reach a high imitation success rate and MPJPE (spherical: 100% Succ and 28.1 $E_{\text{g-mpjpe}}$, VQ: 99.8 % Succ and 36.5 $E_{\text{g-mpjpe}}$), both lose

the ability to serve as a generative model: random samples from the latent space do not generate coherent motion. In NPC, an additional prior needs to be learned. The quantized latent space also introduces artifacts, such as high-frequency jitter, since the network is switching between discrete codes. We visualize this artifact in our `supplement site`'s last section.

## C.4   DOWNSTREAM TASKS

Each generative downstream task policy $\pi_{\text{task}}$ is a three-layer MLP with units [2048, 1024, 512]. For VR controller tracking, we use a six-layer MLP of units [2048, 1536, 1024, 1024, 512, 512]. The value function has the same architecture as the policy. All tasks are optimized using PPO. For simpler tasks (speed, reach, strike), we train for $\sim 2 \times 10^9$ samples. For the complex terrain traversal task, the policy converges after $\sim 1 \times 10^{10}$ samples. The strike, speed, and reach tasks follow the definition in ASE (Peng et al., 2022), while the following trajectory task follows PACER Rempe et al. (2023). VR controller tracking task follows QuestSim Winkler et al. (2022).

**Speed**. For training the x-direction speed task, the random speed target is sampled between 0 m/s $\sim$ 5m/s (the maximum target speed for running in AMASS is around 5m/s). The goal state is defined as $\boldsymbol{s}_t^{\text{g-speed}} \triangleq (d_t, v_t))$ where $d_t$ is the target direction and $v_t$ is the linear velocity the policy should achieve at timestep t. The reward is defined as $r_{\text{speed}} = \text{abs}(v_t - \boldsymbol{v}_t^0)$ where $\boldsymbol{v}_t^0$ is the humanoid's root velocity.

**Strike**. For strike, since we do not have a sword, we substitute it with "strike with hands". The objective is to knock over the target object and is terminated if any body part other than the right hand makes contact with the target. The goal state $\boldsymbol{s}_t^{\text{g-strike}} \triangleq (\boldsymbol{x}_t, \dot{\boldsymbol{x}}_t))$ contains the position and orientation $\boldsymbol{x}_t$ as well as the linear and angular velocity $\dot{\boldsymbol{x}}_t)$ of the target object in the agent frame. The reward is $r_{\text{strike}} = 1 - \boldsymbol{u}^{\text{up}} \cdot \boldsymbol{u}_t$ where $\boldsymbol{u}^{\text{up}}$ is the global up vector and $\boldsymbol{u}_t$ is the target's up vector.

**Reach**. For the reach task, a 3D point $\boldsymbol{c}_t$ is sampled from a 2-meter box centered at (0, 0, 1), and the goal state is $\boldsymbol{s}_t^{\text{g-reach}} \triangleq (\boldsymbol{c}_t))$. The reward for reaching is the difference between the humanoid's right hand and the desired point position $r_{\text{reach}} = \exp(-5\|\boldsymbol{p}_t^{\text{right hand}} - \boldsymbol{c}_t\|_2^2)$.

**Trajectory Following on Complex Terrains** .  The humanoid trajectory following on complex terrain task, used in PACER (Rempe et al., 2023), involves controlling a humanoid to follow random trajectories through stairs, slopes, uneven surfaces (Rudin et al., 2021), and to avoid obstacles. We follow the setup in P and train policy $\pi_{\text{task}}(\boldsymbol{z}_t|\boldsymbol{s}_t^{\text{p}}, \boldsymbol{s}_t^{\text{g-terrain}})$, $\boldsymbol{s}_t^{\text{g-terrain}} \triangleq (\boldsymbol{o}_t, \boldsymbol{\tau}_{1:T})$ where $\boldsymbol{o}_t$ represents the height map of the humanoid's surrounding, and $\boldsymbol{\tau}_{t+10}$ is the next 10 time-step's 2D trajectory to follow. The reward is computed as $r_t^{\text{terrain}} = \exp(-2\|\boldsymbol{p}_t^{(0)} - \boldsymbol{\tau}_t\|) - 0.0005 \cdot \sum_{j \in \text{joints}} |\boldsymbol{\mu}_j \dot{\boldsymbol{q}}_j|^2$ where the first term is the trajectory following the reward and the second term an energy penalty. Random trajectories are generated procedurally, with a velocity between [0, 3]m/s and acceleration between [0, 2] m/s². The height map $\boldsymbol{o}_t$ is a rasterized local height map of size $\boldsymbol{o}_t \in \mathcal{R}^{32 \times 32 \times 3}$, which captures a 2m $\times$ 2m square centered at the humanoid. We do not consider any shape variation or human-to-human interaction as in PACER. Different from PACER, which relies on an additional adversarial reward to achieve realistic and human-like behavior, our framework policy does not rely on any additional reward but can still solve this challenging task with human-like movements. We hypothesize that this is a result of sampling from the pre-learned prior, where human-like motor skills are easier to sample than unnatural ones.

**VR Controller Tracking**. Tracking VR controllers is the task of inferring full-body human motion from the three 6DOF poses provided by the VR controllers (headset and two hand controllers). Following QuestSim (Winkler et al., 2022), we train this tracking policy using synthetic data. Essentially, we treat the humanoid's head and hand positions as a proxy for headset and controller positions. One can view the VR controller tracking task as an imitation task, but with only three joints to track, with the goal state being: $\boldsymbol{s}_t^{\text{g-vr}} \triangleq (\hat{\boldsymbol{\theta}}_{t+1}^{\text{vr}} \ominus \boldsymbol{\theta}_t^{\text{vr}}, \hat{\boldsymbol{p}}_{t+1}^{\text{vr}} - \boldsymbol{p}_t^{\text{vr}}, \hat{\boldsymbol{v}}_{t+1}^{\text{vr}} - \boldsymbol{v}_t, \hat{\boldsymbol{\omega}}_t^{\text{vr}} - \boldsymbol{\omega}_t^{\text{vr}}, \hat{\boldsymbol{\theta}}_{t+1}^{\text{vr}}, \hat{\boldsymbol{p}}_{t+1}^{\text{vr}})$ where the superscript $^{\text{vr}}$ refers to selecting only the head and two hands joints. During training, we use the same (full-body) imitation reward to train the policy. We use the same progressive training procedure for training the tracking policy.

