# OpenReview forum: "Universal Humanoid Motion Representations for Physics-Based Control"
_ICLR.cc/2024/Conference — ICLR 2024 spotlight_

### Official Review · Reviewer_arMx · 2023-11-01

**Soundness:** 3 good
**Presentation:** 3 good
**Contribution:** 3 good
**Rating:** 8
**Confidence:** 4

**Summary:**

This work provides a method to learn a universal humanoid motion prior for different downstream tasks, the prior is designed to cover a wide range of motions. The learned latent space can be used for long, stable and diverse motion generation, and also for solving tasks with natural motions. The latent is learnt by first training an imitator controller to imitate a wide range of human motions, and then learn the latent space by distilling the learned motion imitator. Results show the learned controller could generate a wide range of human motion, and outperform baselines on downstream tasks by a large margin.

**Strengths:**

1. The work is well-motivated, addressing the problem that motion imitator can hardly be used for downstream tasks (especially when interaction with the environment is required) and learned motion prior can not cover a wide range of human motions.
2. Method is well demonstrated with details, and most designs are well-explained. Clear ablation study also shows the effectiveness of different components. Additional implementation details and hyper-parameters are provided in the supplementary materials for the community to reproduce the results.
3. The learned humanoid motion prior showed impressive motion imitating performance over a wide range of human motion, and also showed promising results when applying to diverse downstream tasks, including motion tracking and locomotion over complex terrains, et al.
4. The supplement videos make the difference between proposed method and baseline more comprehensive.

**Weaknesses:**

1. One of this work’s claims is that previous work has limited coverage of the learned latent space that can not cover the wide spectrum of possible human motion. But in Table 1, there is no comparison with ASE or CALM or Imitate&Repurpose on motion imitation performance. Though ASE, or CALM might be a bit hard to compare, Imitate&Repurpose should be reasonable to compare with. I’m not expecting it to perform better, just for completeness.
2. Though PHC+ exhibit great performance on motion imitation, it’s comparison with PHC might be a bit unfair, since one of the modification fo PHC+ is “removing” some hard-negative in the dataset, provide motion imitation result on modified dataset might be beneficial.
3. Some content is a bit hard to read: In Figure 3, it’s quite hard to see the human motion in the Figure 3(e) row, and the title for each row is really hard to see. (Minor issue)

**Questions:**

1. Is the proposed method robust against methodology and dynamics changes? It would be interesting to see these results and potentially enable appling the proposed method to humanoid robots.

---

> ### Author Response · Authors · 2023-11-20
> **Author Response to Review arMx**
>
> We thank the reviewer for helpful, constructive, and positive feedback. We are glad that you find our work "well-motivated", "well-demonstrated", and "outperform baselines on downstream tasks by a large margin". Here, we try to address the questions and concerns:
>
>
> **Comparison with Baselines on Imitation**
>
> As ASE and CALM do not use a motion-tracking objective for training, we cannot compare motion imitation results directly. We show that they do not have good motion skill coverage in the VR controller tracking task, which is a generalization and arguably harder task than full-body motion imitation. The VR controller tracking task only has partial body observations, but requires the policies to produce the same full-body motion as the full-body imitation task. Between PULSE and imitate \& repurpose, the main difference is how the latent space is trained, where PULSE is through distillation and imitate \& repurpose is through RL. Imitate \& repurpose also does not use a Gaussian prior or learnable prior, though both PULSE and Imitate \& repurpose use the AR1 prior. We report the result of training using RL and no distillation in Table 6 of Appendix C.2. We can see that training without distillation can learn a large amount of motion in AMASS (76\%), but it cannot scale to the performance of PULSE. We hypothesize that random sampling for variational bottleneck and random sampling for RL hampers the learning process. This ablation can be viewed as imitate \& repurpose implemented in our framework and humanoid, and we show that it does not scale as well in the imitation task.
>
> **Comparison between PHC and PHC+**
>
> Both PHC and PHC+ are evaluated on the same set of modified training and testing sequences in Table 1.  After removing sequences with issues, there are 11313 and 138 AMASS motion sequences for training and testing, and we re-run all PHC's evaluations on the modified dataset.
>
> **Plot Content Visibility**
>
> We have updated the plot for better visibility. Thanks!
>
> **Answer to Questions**
>
> - As sim2real from physics simulation [1, 2, 3, 4] continues work, we believe that a similar methodology should transfer to real humanoids. Although modifications to the humanoid kinematic tree, reward designs, and domain randomization will be required, the methodology should be applicable. PHC has also demonstrated motion imitation on diverse body shapes, which could also be distilled to a shape/dynamics-aware latent space.
>
>
>
> [1] Cheng, Xuxin, Kexin Shi, Ananye Agarwal, and Deepak Pathak. 2023. “Extreme Parkour with Legged Robots.” arXiv [Cs.RO]. arXiv. http://arxiv.org/abs/2309.14341.
>
> [2] Zhuang, Ziwen, Zipeng Fu, Jianren Wang, Christopher Atkeson, Soeren Schwertfeger, Chelsea Finn, and Hang Zhao. 2023. “Robot Parkour Learning.” arXiv [Cs.RO]. arXiv. http://arxiv.org/abs/2309.05665.
>
> [3] Fu, Zipeng, Ashish Kumar, Ananye Agarwal, Haozhi Qi, Jitendra Malik, and Deepak Pathak. 2021. “Coupling Vision and Proprioception for Navigation of Legged Robots.” http://arxiv.org/abs/2112.02094.
>
> [4] Radosavovic, Ilija, Tete Xiao, Bike Zhang, Trevor Darrell, Jitendra Malik, and Koushil Sreenath. 2023. “Learning Humanoid Locomotion with Transformers.” arXiv [Cs.RO], March. https://doi.org/10.48550/ARXIV.2303.03381.

---

> > ### Comment · Reviewer_arMx · 2023-11-21
> > **Response**
> >
> > Thank you authors for addressing the questions.
> >
> > I'll keep my original evaluation.

---

> ### Author Response · Authors · 2023-11-22
> **Thanks for your suggestions!**
>
> Thanks again for your questions and discussion. It will help us improve the paper and mention the pure RL-based experiments in the main paper.
>
> We appreciate the discussion and constructive comments.
>
> Best,
>
> Authors

---

### Official Review · Reviewer_UBzx · 2023-11-02

**Soundness:** 3 good
**Presentation:** 3 good
**Contribution:** 3 good
**Rating:** 8
**Confidence:** 2

**Summary:**

In the context of human motion representation, the paper proposes a method to create a fundamental representation of humanoid motion that can be used for humanoid control, human motion generation or motion tracking. This representation is created via two main elements: an imitation method and a physics-based learned prior.
The distillation is made through a VAE-like architecture that learns the prior R and decoder D that are then used for each downstream task to generate to generate the action.

**Strengths:**

- The paper is well written
- The presented goal of learning a universal human motion latent representation is very interesting, and the effectiveness on relevant downstream tasks is well presented
- The curation of the mocap training dataset for PHC (along with other modifications) increases robustness and allows fail states recovery
- The downstream tasks are relevant and show interesting use cases for the learned representation
- The ablation study is quite compelling

**Weaknesses:**

- Quantitative results on the VR-controller tracking task are a little bit disappointing compared to Scratch
- The references section could be cleaned up and harmonised, notably for publication conferences
- Although the writing is clear, some typos remain (e.g. Guassian)

**Questions:**

- In section 4.3, a failure case is described using the prior R but it is unclear to me how do we recover from it apart from restarting the task's policy learning? Does it happen often?

---

> ### Author Response · Authors · 2023-11-20
> **Author Response to Review UBzx**
>
> The authors thank the reviewer for constructive and encouraging feedback. We are glad that you find our method a "fundamental representation of humanoid motion", our method's effectiveness "well presented", and ablation "compelling". To address questions and concerns:
>
> **VR Motion Tracking Results**
>
> We acknowledge that the quantitative result on VR motion tracking is less performant on joint position errors though better on success rate. Similar performance degradation on joint errors can also be observed in Table 1, where after distillation with the variational latent space, the imitation performance is worse than the teacher's. This is a known trade-off between prior regularization and achieving good reconstruction performance in VAEs, where the reconstruction error usually does not reach 0 during training. As shown in Appendix C.3, if a non-variational latent space is used (e.g. spherical), a better distillation performance can be achieved (100\% success rate and 28.1 $E\_\text{mpjpe}$). However, such a latent space does not provide a variational prior, and random samples from it do not generate coherent motion. Thus, while our latent space has good coverage of the motor skills required to perform AMASS, it is not full coverage. Downstream task such as VR motion tracking is searching in a constrained solution space and can lose performance on precise motion tracking (though gains faster convergence). Future investigation is needed to achieve better motion imitation results while retaining the generative capabilities of the latent space.
>
>
> **Failure Case not Using Prior R**
>
> We believe that this refers to the cases "one can also supervise $\pi\_{\text{task}}$'s distribution using prior $\mathcal{R}$", which leads to an "adverse feedback loop". We apologize for the confusing statement. Here, we are referring to an alternative to using the residual action formulation. In Sec. 4.3,  we propose to let downstream policy produce a residual with respect to the learned prior $\boldsymbol{a}\_t = \mathcal{D}(\pi\_{\text{task}}(\boldsymbol{z}\_t | \boldsymbol{s}^{p}\_t, \boldsymbol{s}^{g}\_t) + \boldsymbol{\mu}^{p}\_t)$. The alternative is to directly output the latent code rather than the residual, and supervise the output distribution with the prior $\mathcal{R}$. We **do not use** this alternative due to the described issue. The paragraph is included as a justification for the residual action formulation. This failure case only applies to the alternative approach and does not apply to the residual action one.
>
>
> The typos and the reference section are cleaned up in the revision. Thanks!

---

> > ### Comment · Reviewer_UBzx · 2023-11-22
> > **Response**
> >
> > I thank the authors for their response that cleared up the misunderstandings that I had reading the paper. My grade stays the same as 8-Accept.

---

> > > ### Author Response · Authors · 2023-11-23
> > > **Thanks for your advice!**
> > >
> > > The authors would like to thank your again for your concerns and discussion. It will help us improve the paper and will encourage us to find a better balance between motion imitation quality and the latent space prior.
> > >
> > > We appreciate the discussion and suggestions.
> > >
> > > Best,
> > >
> > > Authors

---

### Official Review · Reviewer_a8Fw · 2023-11-03

**Soundness:** 3 good
**Presentation:** 3 good
**Contribution:** 3 good
**Rating:** 8
**Confidence:** 3

**Summary:**

This paper presents a system to train human motion controllers that can imitate large motion datasets and be used efficiently for training tasks such as terrain navigation and vr motion following.

Key contributions:

1. Modification of PHC to learn a better motion controller.

2. A VAE like distillation process to obtain a controller and an action space that can be used efficiently for downstream tasks.

**Strengths:**

1. A controller that can imitate the whole AMASS dataset.

2. A wide range of tasks to verify the efficiency of the system.

**Weaknesses:**

1. No comparison with baselines that resemble the proposed method, e.g., Physics-based character controllers using
conditional VAEs, by Won et al or ControlVAE: Model-based learning of generative controllers for physics-based characters. by Yao et al, which also uses VAEs.

2. The main difference between the proposed system and other systems is that the proposed system is able to scale to the whole AMASS dataset while other systems are mainly doing locomotion or something similar. However, this is not well demonstrated in the downstream tasks, which are mostly just comprised of locomotion tasks in addition to some simple reaching, which the other systems can already do pretty well (maybe less efficiently?).

**Questions:**

1. It will be nice to showcase some scenarios where the benefit of learning the whole AMASS dataset is useful.

2.  What are the motions that PHC cannot handle?

3. I feel like the motion quality produced in the downstream tasks is suboptimal/unnatural. It will be nice to have a metric to measure the motion quality generated in the downstream task for potential future improvement for future work.

4. For the speed task (and other tasks as well, but speed task is one of the results in the ASE paper, so I will focus on this), looks like the motion quality of ASE is really bad, while the original ASE paper has pretty good motion quality (at least visually), any comment on the discrepancy?

---

> ### Author Response · Authors · 2023-11-20
> **Author Response to Review a8Fw 1/2**
>
> We thank the reviewer for the positive feedback and constructive comments. To address the questions and concerns:
>
> **Comparison PhysicsVAE and ControlVAE**
>
> We acknowledge that PhysicsVAE [1] and ControlVAE [2] and ours all strive to learn a compact latent space using a variational bottleneck. While our structure resembles a VAE, we do not use auto-encoding/reconstruction as a learning objective. Our decoder outputs actions $\boldsymbol{a}\_t$ instead of the input state $\boldsymbol{s}\_t$. As a result, we do not require learning any world model during training. This design allows us to directly distill motor skills from a pretrained imitator that has achieved strong performance on a large-scale (40 hours) motion dataset.
>
> PhysicsVAE[1] has tested on the CMU dataset (a subset of AMASS) to further understand its scalability. They "observed that the model learned from the CMU dataset does not perform well when compared to other models learned from either the locomotion dataset or the dancing dataset" (Sec.6, third paragraph, page 11 in PhysicsVAE). The learned latent space from PhysicsVAE also requires an additional adaption layer to solve some of the harder tasks such as terrain traversal, which PULSE does not require.
>
> Similarly, ControlVAE [2] stated that "experiment on a large-scale dataset composed of 3.3 hours of motions" "converges to a lower reward", and the "learned skill embedding can still recover input motion with a larger visual discrepancy, but the performance on the downstream task are significance degrenrated" (Sec.5.7, last paragraph, page 14 in PhysicsVAE).
>
> Compared to PhysicsVAE and ControlVAE, we learn from a much larger dataset and demonstrate that our latent space can (1) scale to the motion dataset after distillation, (2) perform well for downstream tasks without any additional adaptations. PhysicsVAE and ControlVAE also use different humanoids and simulation environments (Bullet and ODE) from ours (SMPL humanoid and Isaac Gym), and the gap between environment and humanoid choices hinders the reimplementation and fair benchmarking.
>
> **Benefit of scaling to the AMASS dataset**
>
> Our main motivation for scaling to AMASS is to have a latent space that covers a wide range of human motion. Thus, we include the VR controller tracking task, which requires tracking three 6DOF points in the global space. This is a generalized version of the full-body motion tracking task, where only partial observation is given to the humanoid to create full-body motion. This task requires the latent space to contain motor skills to match **arbitrary** user input in a continuous and streaming fashion. A latent space that has only seen small-scale locomotion datasets would not have the motor skills required to perform these free-form motions. Our supplement video (Sec. Motion Tracking Downstream Task) shows that the controller that uses our latent space can handle motions such as squatting, punching, dancing, running, rolling, etc. We also quantitatively demonstrate the tracking performance on the AMASS dataset in Table 2. Notice that the joint errors reported in Table 2 are **full-body** joint errors, and a low value means that the full-body motion generated from three-point input is close to the ground truth. Table 2 and Fig. 5 show that using our latent space, one can efficiently solve this task and achieve a high success rate on training, testing, and real-world data capture. Previously proposed latent spaces (ASE and CALM), when also trained with the AMASS dataset, do not obtain the coverage demonstrated in our latent space. We also note that few prior latent space methods have attempted a motion-tracking task due to the diversity and coverage required for free-form motion tracking.
>
> On generative tasks, we test simple ones ranging from reach, speed, and strike, but also challenging ones like complex terrain traversal. The terrain traversal task requires the humanoid to handle stairs, slopes, rough surfaces, and obstacles. Prior art  [3]  on this task has relied on GAIL to achieve natural motion. Using our latent space, we no longer require GAIL but can still learn human-like behavior. Note that previous latent space approaches either fail on a stairs-only environment [4] or require an adaptation layer to traverse rough terrains/slopes [1], while our latent space can be used to solve the task without modification, thanks to the wider range of motor skills it can draw upon.

---

> ### Author Response · Authors · 2023-11-20
> **Author Response to Review a8Fw 2/2**
>
> **Answers to Questions**
>
> - We design the VR motion tracking task to show that our learned latent space can cover a large amount of motor skills. A controller using our latent space is able to achieve good full-body motion tracking performance on the AMASS train and test and also transfers well to real-world data captures using commercial VR headsets. If the latent space has only been trained on a handful of locomotion sequences, it will only have the skills to perform simple actions. Our result on complex terrain traversal also shows that scaling matters, where the humanoid will employ skills like jumping when traveling fast on stairs.
>
> - We will include failure cases for PHC+ from the AMASS test set and other data sources. In general, while PHC can imitate a large number of motion sequences, unseen highly dynamic ones (e.g. badminton jump smash) or noise motion estimated from videos can still cause the humanoid to fall.
>
>
>
> - We agree that motion naturalness can be hard to evaluate and subjective, and we included a large number of video results for qualitative evaluation. The motion quality of a generative method is hard to quantify since there is no ground truth to compare against. Thus, we follow the prior art in the field and provide reward comparisons and video demonstrations. For the VR tracking task, we include pose estimation metrics, which are quantifiable ways to judge the quality of estimated motion. For unnaturalness in motion, we acknowledge that PULSE does not strictly constrain itself to certain types of motion when solving tasks. Since we train on a large-scale motion dataset, certain styles of motion that may be deemed unnatural for the speed task are also available in the latent space. For instance,  AMASS contains motions for "running fanatically" or "limping", whose motor skills are also learned in the latent space and can be used to solve the speed task. While training from scratch will often result in inhuman behavior, PULSE solves tasks using human-like behavior even when there are no constraints on the strategies used. To further constrain the solution space to natural behavior defined for certain tasks, one can enforce symmetry or incorporate additional regularizations.
>
>
> - The discrepancy on baselines (ASE and CALM) mostly comes from the training data scale. For a fair comparison, all baselines are trained on the AMASS training set, the same as PULSE. The original ASE is trained on a specialized dataset that only contains locomotion or character movement (eg. strike). By construction, a latent space trained on specialized datasets will only include motor skills that are deemed natural for the downstream task. However, datasets like AMASS contain diverse full-body motion that are not specialized. Thus, motor skills for sequences such as "crouch walking" or "running fanatically" could also be learned in the latent space of ASE. When using these motor skills for downstream tasks, the resultant motion can appear unnatural, though still human-like (in the sense that a human can recreate these motions if desired).
>
>
> [1] Won, Jungdam et al. “Physics-based character controllers using conditional VAEs.” ACM Transactions on Graphics (TOG) 41 (2022): 1 - 12.
>
> [2] Yao, Heyuan, Zhenhua Song, Baoquan Chen, and Libin Liu. 2022. “ControlVAE: Model-Based Learning of Generative Controllers for Physics-Based Characters.” arXiv [Cs.GR]. arXiv. http://arxiv.org/abs/2210.06063.
>
> [3] Rempe, Davis, Zhengyi Luo, Xue Bin Peng, Ye Yuan, Kris Kitani, Karsten Kreis, Sanja Fidler, and Or Litany. 2023. “Trace and Pace: Controllable Pedestrian Animation via Guided Trajectory Diffusion.” arXiv [Cs.CV]. arXiv. http://arxiv.org/abs/2304.01893.
>
> [4] Hasenclever, Leonard, Fabio Pardo, Raia Hadsell, Nicolas Heess, and Josh Merel. 13--18 Jul 2020. “CoMIc: Complementary Task Learning \& Mimicry for Reusable Skills.” In Proceedings of the 37th International Conference on Machine Learning, edited by Hal Daumé Iii and Aarti Singh, 119:4105–15. Proceedings of Machine Learning Research. PMLR.

---

> ### Author Response · Authors · 2023-11-21
> **Additional Experiment for showcasing the benefit of scaling to AMASS**
>
> We would like to thank the reviewer again for raising the concern over the benefit of scaling to AMASS. Here, we would like to add some quantitative evidence.
>
> We study the effect of training our framework on a locomotion-only dataset (a subset of AMASS, used in PACER [3]) that contains 275 motion sequences of a total length of 38 minutes. The dataset contains mainly walking, running, and jumping. After distilling, the encoder-decoder imitator can reach an imitation result of:
>
>   | AMASS-Locomotion |                                  |                                  |                                  |                                  |
> | ----------- |----------- |----------- |----------- |----------- |
>   | $Succ \uparrow$       | $E\_{g-mpjpe} \downarrow$        | $E\_{mpjpe} \downarrow$        | $E\_{acc} \downarrow$        | $E\_{vel} \downarrow$       |
> |       100%                           |  25.5                                |     20.1                             |                 2.9                 |          4.3  |
>
> which shows that the latent space has successfully learned the training sequences. Random sampling from the latent space also results in human-like walking and running motion.
>
> However, when applying this latent space  to the VR controller tracking task, we obtain the following result on the AMASS train and test, as well as the real-world data capture:
>
>  | | AMASS-Train |                                  |                                  |                                  |                                  |  AMASS-Test |                                  |                                  |                                  |                                  |
> | ----------- | ----------- |----------- |----------- |----------- |----------- | ----------- |----------- |----------- |----------- |----------- |
> | | $Succ \uparrow$   | $E\_{g-mpjpe} \downarrow$        | $E\_{mpjpe} \downarrow$        | $E\_{acc} \downarrow$        | $E\_{vel} \downarrow$       |  $Succ \uparrow$       | $E\_{g-mpjpe} \downarrow$        | $E\_{mpjpe} \downarrow$        | $E\_{acc} \downarrow$        | $E\_{vel} \downarrow$     |
> | Locomotion-latent |  56.8 % |   111.6 | 75.4 | 4.5 | 8.8 |  15.2 % |   138.2 | 90.1 | **5.7** | **10.9** |
> | Ours | **99.5 %** |   **57.8** | **51.0** | **3.9** | **7.1** |  **93.4 %** |   **88.6** | **67.7** | 9.1 | 14.9 |
>
>
>  | |Real-World |                                  |                                  |                                  |                                  |                         |
> | -----------| ----------- |----------- |----------- |----------- |----------- |----------- |
> |  | $Succ \uparrow$       | $E\_{g-mhpe} \downarrow$         | $E\_{acc} \downarrow$        | $E\_{vel} \downarrow$       |
> | Locomotion-latent |5/14 | 117.5 | **5.5** | 9.5 |
> | Ours |**14/14** | **68.4** | 5.8 | **9.0** |
>
> Both ours and the locomotion-latent model have the same training data and procedure, with the only difference being the training data when forming the latent space. From the result we can see that scaling to AMASS significantly increases the VR controller tracking task's performance on the real-world dataset, showcasing its usability in real-world scenarios. Scaling to a large dataset increases the diversity and coverage of the latent space and improves its applicability to downstream tasks.
>
> We would love to further discuss and provide explanations if the reviewer has any additional concerns or questions. Thanks!

---

> > ### Comment · Reviewer_a8Fw · 2023-11-21
> > **response**
> >
> > Thanks for clarifying my questions. I raise my score accordingly.

---

> ### Author Response · Authors · 2023-11-22
> **Thanks for your constructive comments!**
>
> Thanks again for your constructive comments and discussion, which will help improve our paper and our presentation on the importance of scaling up to a bigger motion dataset.
>
> We appreciate the discussion and raising the score.
>
> Best,
>
> Authors

---

### Author Response · Authors · 2023-11-20
**General Responses and Revision Summary**

We thank the reviewers for their time and constructive feedback and hope that our response could address some of the concerns. We are glad that they find our work to create a "fundamental representation for humanoid motion" (UBzx), "efficient" (a8Fw), "well-ablated" (UBzx, arMx), and "outperform baselines on downstream tasks by a large margin" (arMx). Indeed, we aim to "address the problem that motion imitator can hardly be used for downstream tasks (especially when interaction with the environment is required) and learned motion prior can not cover a wide range of human motions." (arMx).

Here we provide a list of revisions we have made based on the constructive feedback (all updates are marked blue in the revised manuscript. ):

- Fixes Sec 4.3 writing description on failure cases.
- Fixed Fig.3 for better visibility
- Updated description of the VR tracking task.
- Typo fixes.
- Added failure cases for PHC+ in supplementary videos [(Supplement site Sec. Failure cases for PHC+)](https://pulse-humanoid.github.io/pulse/#mocap_motion_imitation_fail). Videos are also updated in zip.
- Updated terrain traversal video to better show case stairs [(Supplement site  Sec. Terrain)](https://pulse-humanoid.github.io/pulse/#terrain). Videos are also updated in zip.

We address reviewers' comments in individual replies and would be glad to answer any additional questions & discussions.

---

### Meta-Review · Area_Chair_Eqkn · 2023-12-01

**Metareview:**

Summary: The paper proposes a method for learning a representation of humanoid motion, designed to cover a broad range of motions. The learned latent space can be used for downstream tasks like humanoid control, stable and diverse human motion generation, and motion tracking. The representation is learned in two steps: 1) train an imitator controller from a large, unstructured motion dataset, that can imitate a wide range of human motions; 2) learn the latent space by distilling the learned motion imitator via a VAE-like architecture. Results demonstrate generalization to a wide range of human motions and better performance that baselines on downstream tasks by a large margin.

Strengths:
- The paper is well written, well motivated, and tackles an interesting and important problem [UBzx, arMx]
- The imitation controller is a valiant and impressive contribution [a8Fw]
- The evaluation tasks are relevant and compelling use cases for the learned representations [a8Fw, UBzx, arMx]
- The proposed method is well demonstrated, with a strong ablation study and comprehensive details to aid reproducibility, as well as good illustrative video comparisons to baselines [UBzx, arMx]

Weaknesses:
- There are several baseline comparisons that are lacking that need to either be included or justified (PhysicsVAE, ControlVAE, ASE, CALM) [a8Fw, arMx].
- Reviewers requested general clarifications (what is the benefit of scaling to AMASS, example motions that PHC can't handle, PHC vs PHC+ comparison).
- Reviewers request writing revision to address typos and references.
- Quantitative results on the VR-controller tracking task are not as compelling [UBzx].

Overall, all reviewers vote to accept this paper, and point out weaknesses that for the most part are resolved in the discussion / will be resolved in the camera ready. I agree with the reviewers that this is a solid paper that should be accepted.

**Justification For Why Not Higher Score:**

The justification for this might have more to do with my own lack of calibration for what qualifies as a oral vs spotlight paper. All reviewers recommend 8s, I'd expect to see at least a 10 to recommend a paper as an accept with oral. I would not mind if this paper was bumped up to Accept with oral though.

**Justification For Why Not Lower Score:**

All reviewers agree this paper proposes a compelling method, with strong evaluation tasks, ablations, and good baselines. The weaknesses are mostly minor clarifications and baseline concerns that the rebuttal has for the most part clarified. I believe building a representation for such a vast array of humanoid motion is incredibly important and highly topical for the current "is scale all we need?" debate in robotics. I believe this paper should at least get a spotlight.

---

### Decision · Program_Chairs · 2024-01-16

Accept (spotlight)